# Spatial Distribution Patterns and Assembly Processes of Abundant and Rare Fungal Communities in *Pinus sylvestris* var. *mongolica* Forests

**DOI:** 10.3390/microorganisms12050977

**Published:** 2024-05-13

**Authors:** Reyila Mumin, Dan-Dan Wang, Wen Zhao, Kai-Chuan Huang, Jun-Ning Li, Yi-Fei Sun, Bao-Kai Cui

**Affiliations:** State Key Laboratory of Efficient Production of Forest Resources, School of Ecology and Nature Conservation, Beijing Forestry University, Beijing 100083, China; ramilla@163.com (R.M.); wdd1360303992@163.com (D.-D.W.); zhaowendrlw@163.com (W.Z.); huangkc1014@bjfu.edu.cn (K.-C.H.); lijunning17@mails.ucas.ac.cn (J.-N.L.)

**Keywords:** *Pinus sylvestris* var. *mongolica*, soil fungi, spatial distribution, assembly processes

## Abstract

Revealing the biogeography and community assembly mechanisms of soil microorganisms is crucial in comprehending the diversity and maintenance of *Pinus sylvestris* var. *mongolica* forests. Here, we used high-throughput sequencing techniques and null model analysis to explore the distribution patterns and assembly processes of abundant, rare, and total fungal communities in *P. sylvestris* var. *mongolica* forests based on a large-scale soil survey across northern China. Compared to the abundant and total taxa, the diversity and composition of rare taxa were found to be more strongly influenced by regional changes and environmental factors. At the level of class, abundant and total taxa were dominated by Agaricomycetes and Leotiomycetes, while Agaricomycetes and Sordariomycetes were dominant in the rare taxa. In the functional guilds, symbiotrophic fungi were advantaged in the abundant and total taxa, and saprotrophic fungi were advantaged in the rare taxa. The null model revealed that the abundant, rare, and total taxa were mainly governed by stochastic processes. However, rare taxa were more influenced by deterministic processes. Precipitation and temperature were the key drivers in regulating the balance between stochastic and deterministic processes. This study provides new insights into both the biogeographical patterns and assembly processes of soil fungi in *P. sylvestris* var. *mongolica* forests.

## 1. Introduction

Soil microorganisms play a crucial role in soil nutrient cycling by decomposing organic matter and releasing nutrients for plant uptake, which contribute to the stability and health of the soil ecosystem [1,2,3,4,5]. Understanding the formation of soil microbial diversity mechanisms is necessary to fully understand their functions in the ecosystem, and it is also one of the key factors of the microbial ecology [6,7,8]. Therefore, recent studies have gradually deepened, to explore microbial internal community assembly mechanisms [9,10]. Generally, two ecological process theories (ecological niche process theory and neutral process theory) together describe the mechanism of microbial community assembly [11,12]. Traditional ecological niche process theory believes that microbial communities are formed by deterministic biological factors (species interaction, such as competition and predation) and abiotic factors (environmental factors, such as pH and temperature) [13,14]. The neutral process theory assumes that the loss and increase of microorganisms in the group shows a random balance that is because of stochastic processes (birth, death, migration, speciation, diffusion limitation), and it gives a specific shape to the microbial community structure [15,16]. At present, these ecological process theories are widely used in relevant research to explore the mechanism of microbial community assembly at the spatiotemporal scale [17,18,19,20]. Although many studies focus on quantifying the relative importance of the stochastic and deterministic processes that are involved in microbial community assembly in different habitats and at different scales (e.g., regional, continental, and global), there is still an ongoing debate on this topic [21,22,23,24,25,26].

In addition, the microbial community usually shows a highly unbalanced pattern of species distribution, consisting of a few highly abundant taxa and a large number of rare taxa [25,27]. Abundant and rare taxa exhibit unequal responses to habitat alteration in terms of their contributions to overall biodiversity and community structure [28]. The different responses of the two taxa may lead to the ecological process of community assembly and maintaining community functions. In previous studies, more research has been conducted on the effects of different habitats on soil microbial communities, but most of the studies had focused only on the total taxa, with little attention paid to abundant and rare taxa. Increasing studies show that abundant and rare taxa have different ecological functions and rare taxa may have more important contributions than abundant taxa [29,30]. Therefore, revealing the different responses of the total, abundant, and rare fungal taxa to environmental changes is the key to understanding the process of microbial community assembly and ecosystem functions [31].

*Pinus sylvestris* var. *mongolica* is one of the important tree species for ecological environment construction in the desert areas of northern China [32,33]. Past research in China mainly focused on the diversity of microbial composition and functional and community structure at local scales such as the Maowusu sandy land, Daxing’an Mountains, Zhanggutai, etc., at different stages of forest development (such as medium forest, near-mature forest, mature forest, etc.), vertical horizontal soil profiles (0–10 cm, 10–20 cm), and different habitats (sandy land, grassland, pure forest, etc.). However, there is a lack of research on dynamics at a regional scale. In particular, the response and community assembly mechanisms of abundant and rare fungal communities in the soil of *P. sylvestris* var. *mongolica* in response to environmental changes are still unknown.

Therefore, *P. sylvestris* var. *mongolica* forests that grow in four different regions of northern China, including Honghuaerji, Yulin, Saihanba, and Zhanggutai, were used as research objects. High-throughput sequencing technology was used to explore the large-scale biogeographical pattern and community assembly mechanism of the abundant, rare and total fungal communities in four *P. sylvestris* var. *mongolica* forests. The purpose of this study is as follows: (1) To explore the spatial distribution patterns and functions of abundant, rare, and total fungal communities in the soil ecosystem of *P. sylvestris* var. *mongolica* forests. (2) To quantify the relative importance of stochastic and deterministic processes, as well as driving factors, of abundant, rare, and total fungal community assembly across different regions. The results of this study will help in understanding how the assembly of abundant, rare, and total fungal communities relates to soil biodiversity and the biogeographical distribution of *P. sylvestris* var. *mongolica*. Additionally, it will provide new insights into analyzing the role of soil microorganisms in maintaining the stability of ecosystem structure and functions.

## 2. Materials and Methods

### 2.1. Study Area and Soil Sampling

In July 2021, soil samples of *P. sylvestris* var. *mongolica* forests were collected from four different regions: Honghuaerji forest park (HB), located in the southeast of Hulunbuir sandy land; Zhanggutai sandy forest park (HQ), located in the southern margin of Horqin sandy land; Yulin sandy forest park (MU), located in the southern edge of Mu Us sandy land; and Saihanba mechanical forest farm (OD), located in the southern edge of Otindag sandy land. Basic information about the four regions is shown in Table 1. Two sampling sites were selected in each region, and within each site, 15 small plots with a size of 20 m × 20 m, and a distance of more than 20 m between each plot, were established. Soil samples were collected from the 0–15 cm soil layer, using the five-point sampling method within each small plot, resulting in one composite sample per plot. A total of 120 soil samples were collected (4 regions × 2 sites × 15 plots). On-site, the soil samples were sieved through a 2 mm sieve and divided into two portions, which were placed in sterile sample bags with corresponding numbers. One portion was stored at −80 °C in a freezer for DNA extraction, while the others were air-dried naturally and used for soil physical and chemical property analyses. These steps were taken to preserve the original characteristics and stability of the soil samples before further analyses in the laboratory [34].

### 2.2. Soil and Meteorological Data

Soil properties were determined using the method described by Bao (2000) [35]. The pH value (pH) of the soil was measured by a pH meter based on a suspended soil in water ratio of 1:2.5. Soil organic carbon (SOC) was determined by the titration of ferrous sulfate heated by potassium dichromate. Total nitrogen (TN) was determined using the Kelvin method, with Se, CuSO_4_, and K_2_SO_4_ as catalysts, and digested with concentrated sulfuric acid using 1 g of soil. MADAC (anti-molybdenum antimony colorimetry) was used to estimate the phosphorus content in soil samples. The cation exchange capacity (CEC) was determined using ammonium acetate exchange by the Kjeldahl method. Climate data for each sampling site were obtained from the China Meteorological Data Service Center (CMDC, http://data.cma.cn/en, accessed on 15 July 2021). Data included the mean temperature (Ta), mean precipitation (Pa), and average maximum and minimum temperature (Tmax and Tmin) of different regions.

### 2.3. Soil DNA Extraction and Amplification

Total genomic DNA was extracted from soil samples using the DNeasy Power Soil Pro Kit (Qiagen, Hilden, Germany). The DNA quantity was determined using a NanoDrop NC2000 spectrophotometer (Thermo Fisher Scientific, Waltham, MA, USA), while the quality of DNA was detected by 1.2% agarose gel electrophoresis. The ITS1 region of the fungi was PCR-amplified with the following primers, ITS1F and ITS2 [36]. The PCR amplification, and all the steps of DNA purification and quantification, were based on the experimental method of Wang et al. (2022) [37]. Amplicons were mixed in proportion to the sequencing amount, and pair-end 2 × 250 bp sequencing was performed using the Illumina MiSeq platform with a NovaSeq 6000 SP reagent kit at Shanghai Personal Biotechnology Co., Ltd. (Shanghai, China).

### 2.4. Sequence Analyses

Microbiome bioinformatics analyses were performed using Qiime2 2020.2, with a slight modification based on the official tutorials. The Demux plugin was utilized to split mixed sequences into individual samples, and the Cutadapt plugin was used to remove primers [38]. The DADA2 plugin was employed for the quality filtering, denoising, and chimera removal of the sequences [39]. After quality screening, non-singleton amplicon sequence variants (ASVs) were aligned using mafft, and a phylogenetic tree was constructed using fasttree2. A taxonomic assignment of the fungal ITS sequences was conducted with the UNITE reference database. Finally, we used the rarefy command of the vegan package in R (R Core Team, 2022) to standardize the ASV table according to the minimum sequence number of samples, and we used it for other subsequent analyses.

### 2.5. Defining Abundant and Rare Taxa

The definitions of “abundant” or “rare” taxa were based on the relative abundances of each ASV [30]. The ASVs with relative abundances > 0.1% of the total sequences were defined as “abundant” taxa, those with relative abundances < 0.01% were defined as “rare” taxa, and those with relative abundances ≥ 0.01% and ≤0.1% were “moderate” taxa [25,40,41].

### 2.6. Null Model Analyses

A null model analysis was performed, using a framework described by Stegen et al. (2013), to classify community pairs into underlying drivers of deterministic processes or stochastic processes [42]. Briefly, variations in phylogenetic and taxonomic diversity were measured using null model-based phylogenetic and taxonomic β-diversity metrics, namely, the β-nearest taxon index (βNTI) and the Bray–Curtis-based Raup–Crick (RCbray). Deterministic selection processes can be inferred when the βNTI is less than −2 (homogeneous selection) or greater than 2 (variable selection). Stochastic processes include dispersal limitation (|βNTI| < 2, RCbray > 0.95) and homogenizing dispersal (|βNTI| < 2, RCbray < −0.95). If the |βNTI| < 2 and |RCbray| < 0.95, the community assembly process is controlled by undominated processes.

### 2.7. Statistical Analyses

Sequence data analyses were mainly performed based on R packages (version 4.1.2) and the online charting website https://www.genescloud.cn/chart/ChartOverview, accessed on 14 July 2023. In an α-diversity analysis, the Chao1 richness index, Shannon diversity index and Pielou evenness index were selected for one-way analysis of variance (ANOVA). Nonmetric multidimensional scaling (NMDS) was performed to test differences in microbial communities in different locations based on the unweighted unifrac distance. A redundancy analysis (RDA) and Mantel test were used to explain the relationship between the environmental factors and microbial composition. NMDS and RDA were performed in R. ANOVA; Mantel tests, functional guilds, and trophic modes were obtained from the charting website and FUNGuild database. The R (version 4.1.2) software “picante” toolkit was used for the null model analysis.

## 3. Results

### 3.1. Diversity of Abundant and Rare Taxa

After quality filtering, removing chimeric sequences, and rarefying, 4,856,520 high-quality sequences were clustered into 2850 ASVs based on a 100% similarity. Across those fungal ASVs, a total of 166 ASVs (5.8%) with 3,589,112 sequences (73.9%) were identified as abundant taxa, while 2107 ASVs (73.9%) with 344,539 sequences (7.1%) were identified as rare taxa, and 577 ASVs (20.3%) with 922,869 sequences (19%) were considered as moderate taxa (Table 2).

A one-way analysis of variance was conducted to compare the α-diversity indices of the different abundance of fungal taxa in *P. sylvestris* var. *mongolica* forests across different regions (Figure 1). According to the range of values, the Chao1 index exhibited a trend of total taxa > rare taxa > abundant taxa, while the Shannon and Pielou indices showed a trend of rare taxa > total taxa > abundant taxa. This suggested that rare taxa contribute significantly to the fungal diversity in the soil of *P. sylvestris* var. *mongolica* forests.

Furthermore, the α-diversity indices of different fungal taxa significantly varied between different sampling sites. Specifically, the highest α-diversity (Chao1 index, Shannon index, Pielou index) was observed in the HQ forest for all fungal groups. The Chao1 indices of abundant and rare taxa showed no significant difference between the HQ and OD sampling sites (*p* > 0.05), while significant differences were observed between other sampling sites (*p* < 0.01). The Chao1 index of the total taxa showed no significant difference between the HB and MU sampling sites (*p* > 0.05), but significant differences were found between other sampling sites (*p* < 0.01). The Shannon and Pielou indices of the different abundance of fungal taxa differed significantly between HB and all other regions (*p* < 0.01).

Differences in soil fungal communities between the four regions were analyzed, as shown in Figure 2. The soil samples from the four regions exhibited a scattered distribution on the NMDS plot (Stress < 0.2), indicating that the fungal community structure in *P. sylvestris* var. *mongolica* forests was influenced by regional factors to some extent. The abundant, rare, and total taxa showed similar community structures between the OD forest and HB forest, while significant differences were observed between the other regions. Furthermore, the NMDS plots of the abundant and total taxa exhibited a high degree of consistency. This suggested that the abundant taxa predominantly shape the fungal community structure in the soil of *P. sylvestris* var. *mongolica* forests. On the other hand, in the NMDS plot of the rare taxa, a large difference can be observed between the rare fungal taxa in the HQ forest and other fungal taxa. This indicates that the distribution of the rare fungal community was more significantly influenced by habitat changes.

### 3.2. The Community Composition and Functional Guilds of Abundant and Rare Taxa

The relative abundances were calculated on the ASV numbers, and they classified fungi with a relative abundance less than 1% as other species and those with a relative abundance greater than 10% as dominant species (Figure 3). The total soil fungal taxa were distributed in 17 phyla, 49 classes, 119 orders, 233 families, and 452 genera in all the sample plots. The distribution of soil fungal communities between the four regions exhibited significant variations. Overall, the dominant phyla in terms of abundant, rare, and total taxa showed similarities, with Basidiomycota (13–76%) and Ascomycota (22–61%) being the primary phyla. However, the relative abundance of these phyla differed significantly across different regions. Furthermore, the dominant classes observed in the fungal taxa with different abundances between the four regions were varied. For the abundant and total taxa, the dominant classes included Agaricomycetes, Leotiomycetes, Eurotiomycetes, and Pezizomycetes (11–76%). The rare taxa were characterized by dominant classes such as Agaricomycetes, Sordariomycetes, Dothideomycetes, and Leotiomycetes (11–35%). At the genus level, the composition of fungal taxa with different abundances was different across the four regions. For instance, the dominant genera were *Inocybe* (23%) and *Cortinarius* (12%) in the HB region; *Inocybe* (13%) and *Geopora* (12%) prevailed in the MU region; *Helvellosebacina* (10%) was the dominant genus for the OD region; and *Pseudogymnoascus* (16%) was dominant in the HQ region. Regarding rare taxa, the majority of genera (87–93%) have a relative abundance of less than 1% in forest regions, and no dominant genera with a relative abundance exceeding 10% were observed. However, there are some relatively common genera found in specific regions: *Inocybe* (8%) and *Tomentella* (4%) in the HB region, *Geopora* (6%) and *Mortierella* (4%) in the MU region, and *Saitozyma* (3%) in the OD and HQ regions.

Significant differences were also found in the major ecological functional groups of different-abundance fungal taxa, based on FUNGuild functional predictions. More multifunctional taxa were predicted in the rare and total taxa, but among them, species for which trophic types could not be identified accounted for the vast majority of the soil fungal composition—52% and 42%, respectively. For the abundant taxa, ectomycorrhizal fungi dominated (62%). This suggests that rare taxa contribute more to multiple functions and trade-offs than abundant taxa, which are more inclined to support a single function. In addition, fungi with ecological functions higher than 2% in the soil were selected for counting, and the results showed that different abundances of fungi contained different amounts of pathogenic, symbiotic, and saprophytic fungi, with the abundant and total taxa being dominated by symbiotic fungi (64% and 18%), and the rare taxa being dominated by saprophytic fungi (17%), which suggests that the type of fungal abundance has a significant impact on the functional guilds of soil fungi.

### 3.3. Assembly Processes of Abundant and Rare Taxa

The null model was used to calculate the phylogenetic turnover rate between each pair of communities. The βNTI and RCbray were employed to quantify the ecological processes driving the assembly of soil fungal communities in *P. sylvestris* var. *mongolica* forests. The relative contributions of deterministic and stochastic processes in community assembly were further determined. The results showed that most βNTI values for different-abundance fungal taxa in various regions were between −2 and 2, indicating that the assembly of soil fungal taxa in *P. sylvestris* var. *mongolica* forests is primarily driven by stochastic processes (94%). The process of community assembly for abundant taxa, versus the total community composition, was generally similar across different regions, with stochastic processes dominating (89–99.5%), and there were no significant differences between different regions (*p* > 0.05). In contrast, rare taxa were predominantly governed by 64% stochastic processes and 36% deterministic processes, and there were significant differences between different regions (*p* < 0.05). For rare taxa, the proportion of stochastic (70% and 59%) and deterministic (20% and 51%) ecological processes differed significantly between the HQ and OD forests. This suggests that, compared to abundant taxa, the assembly of rare taxa is more influenced by spatial distance (Figure 4).

By coupling the βNTI and RCbray data, we divided the community assembly processes into four components. For the overall sampling sites, different-abundance fungal taxa were dominated by different ecological processes in terms of stochasticity. In general, for both the rare and total taxa, different ecological processes played relatively equal roles in driving community assembly, in terms of fungal community turnover. Specifically, homogeneous dispersal, dispersal limitation, and undominated processes accounted for 38.3%, 23.0%, and 32.1% of the turnover for the total taxa, and 42.49%, 30.0%, and 34.59% for the rare taxa, respectively. For abundant taxa, undominated processes (drift, weak selection, weak dispersal, diversification) were the most important ecological processes, followed by homogeneous dispersal, explaining 65.0% and 29.3% of the community turnover, respectively. As for deterministic processes, the total taxa were dominated by 55% homogeneous selection and 45% variable selection. However, abundant and rare taxa were primarily governed by 99% and 97% variable selection, respectively.

### 3.4. The Impact of Environmental Factors on Diversity, Composition, and Assembly Process of Abundant and Rare Taxa

#### 3.4.1. The Impact of Environmental Factors on Diversity of Abundant and Rare Taxa

The impact of environmental factors on the α-diversity of different-abundance fungal communities were further confirmed by Pearson correlation analyses (Table 3). The results showed that different environmental factors contributed differently to the variation of α-diversity indices for different abundances of fungal taxa. For the abundant taxa, Pa (positive) and pH (negative) were strongly correlated with the Chao1 index (*p* < 0.01), while other environmental factors were weakly correlated or not significantly correlated with the α-diversity indices. For the rare taxa, Pa (positive), AP (negative), and Tmin (positive) were strongly correlated with the Chao1 index; AP (negative), Pa (positive), and Tim (positive) with the Shannon index; and Tim (positive) and AP (negative) with the Pielou index (*p* < 0.01); other environmental factors were weakly correlated or not significantly correlated with the α-diversity indices. For the total taxa, Pa (positive), AP (negative), and pH (negative) were strongly correlated with the Chao1 index, and AP (negative) with the Shannon index (*p* < 0.01), while other environmental factors were weakly correlated or not significantly correlated with the α-diversity indices. Thus, it can be seen that most environmental factors contributed more to the Chao1 index than the Shannon and Pielou indices, and that Pa was the most influential environmental factor for all fungal taxa.

A correlation analysis was conducted at the ASV level to identify the environmental factors influencing the soil fungal community composition in *P. sylvestris* var. *mongolica* forests (Figure 5B). The results revealed that the overall response of abundant, rare, and total taxa to environmental factors were generally similar across all regions. However, the intensity of the response to different environmental factors varied between different regions. Soil nutrients (TN, SOC, CEC) were identified as the main environmental factors influencing the fungal community structure in the HB and OD forests. Soil pH and temperature were primary environmental factors influencing fungal community structure in the MU forest. Moisture and temperature were the main environmental factors influencing the fungal community in the HQ forest. In comparison to the abundant taxa, the positions of the regions on the quadrants and the impact of environmental factors on the rare and total fungal community structures were more similar, while stronger correlations were observed in rare taxa. In conclusion, the differences in the abundance and composition of fungal communities in *P. sylvestris* var. *mongolica* forest soil across different regions can largely be attributed to variations in the strength of correlations with different environmental factors.

#### 3.4.2. The Impact of Environmental Factors on Community Compositions of Abundant and Rare Taxa

To further explore the impact of environmental factors on the composition of individual fungal communities at varying abundance levels, we conducted Mantel correlation analyses between a Euclidean distance matrix based on dominant taxonomic classes and various environmental variables (Figure 5A). We focused on the top two dominant classes with the highest relative abundances (an average relative abundance exceeding 10%) and their relationship with the environmental variables. The dominant class Agaricomycetes exhibited significant positive correlations (*p* < 0.01) with Tmax, Tmin, SOC, and TN (*p* < 0.01) among abundant, rare and total taxa. Notably, Agaricomycetes demonstrated the strongest associations with Tmax and Tmin in the abundant and total taxa, while displaying significant correlations (*p* < 0.05) with Pa and SOC in the total taxa. Conversely, within rare taxa, Agaricomycetes showed stronger relationships with soil SOC and TN. The dominant class Leotiomycetes, representing both abundant and total taxa, demonstrated significant positive correlations (*p* < 0.01) with Pa and Tmin during the immature stage. For rare taxa, the dominant class Sordariomycetes exhibited significant positive correlations (*p* < 0.01) with Tmax, Tmin, Pa, and SOC (*p* < 0.05) (Figure 5B). These findings suggest that there are discernible differences in the influencing factors on the composition of fungal taxa at varying abundance levels.

#### 3.4.3. The Impact of Environmental Factors on Assembly Processes of Abundant and Rare Taxa

To explore the primary environmental factors that determine the assembly process of abundant, rare, and total taxa, we performed linear regression analyses between the βNTI values and each environmental variable. The results showed that different environmental factors had varying degrees of impact on the assembly process of different-abundance fungal taxa (Figure 6). Regardless of the fungal taxa, precipitation was the most critical environmental factor, having the strongest positive correlation with the assembly process of abundant taxa, followed by rare taxa (both *p* < 0.01), and a significantly negative correlation with the total taxa (*p* < 0.01). In addition, pH and Tmin had significant impacts on the total taxa, while other environmental factors had relatively low correlation coefficients. Specifically, in response to environmental factors, abundant and rare taxa exhibited similar patterns, and the βNTI values of both abundant and rare taxa were not significantly correlated with pH (*p* > 0.05) but showed significant correlations with other environmental factors (*p* < 0.01). The correlation coefficients for Tmin (positive), AP (negative), (CEC) (negative), SOC (negative), TN (negative), and Tmax (positive) were in descending order. However, compared to abundant taxa, the βNTI values of rare taxa show stronger correlations with various environmental factors.

## 4. Discussion

Forests are complex terrestrial ecosystems that play a crucial role in maintaining the global ecological balance. Understanding the mechanisms underlying the assembly of microbial communities is essential for the generation and maintenance of biodiversity, and it has been extensively explored in the field of microbial ecology [43]. Previous studies have shown that the soil microbial communities of *P. sylvestris* var. *mongolica* forests often exhibit different spatial and seasonal distribution patterns in different sandy lands [44,45]. On a large spatial scale, temperature and precipitation are the main driving factors shaping bacterial and fungal community changes. Additionally, fungal communities are also influenced by pH, leading to distinct geographical distribution variations [30]. However, the geographical distribution patterns and community assembly processes of fungal communities with different abundances in *P. sylvestris* var. *mongolica* forests are currently poorly understood, and the role of the key driving factors involved has remained unclear. Therefore, this study aimed to explore the mechanisms and driving factors behind the assembly of abundant, rare, and total fungal communities in *P. sylvestris* var. *mongolica* forest soils across different regions.

### 4.1. Diversity of Abundant and Rare Taxa and Impacting Factors

Based on a comprehensive analyses of all soil samples, it is consistently observed that the α-diversity of rare taxa in *P. sylvestris* var. *mongolica* forest is higher than that of abundant taxa. This finding aligns with the consensus among other researchers, highlighting the essential role of rare fungal communities as a reservoir of microbial diversity and their significant contribution to the overall maintenance of microbial biodiversity [46,47]. Furthermore, numerous studies have demonstrated that soil microbial communities often exhibit distinct adaptations to diverse soil environments, leading to variations in their survival strategies and physiological mechanisms and resulting in varying levels of diversity in soil microbial communities [37]. For instance, Lin found that there were significant differences in the abundance and diversity of abundant and rare fungal communities in soils of the same tree species in both China and Japan [47]. This difference was thought to be related to the climate, soil, and vegetation types of the two locations. Similarly, Zhou et al. (2017) [48] observed dissimilar proportions of rare and abundant fungal taxa in soils beneath eucalyptus forests across different regions of Australia, highlighting the close relationship between soil fungal communities and factors such as soil pH and organic matter [49]. Our findings are similar to those of previous studies and provide substantial support to their claims. Specifically, our study revealed the significant influence of regional variations on both the α- and β-diversity of different-abundance fungi within *P. sylvestris* var. *mongolica* forests. However, the impact of regional differences on the α-diversity of each fungal group were primarily reflected in the Chao1 index, while the Shannon and Pielou indices displayed negligible effects. This indicates that variations may exist in the species abundance of abundant and rare fungal communities within *P. sylvestris* var. *mongolica* forest soils across various regions. Nonetheless, the overall diversity and evenness of fungal communities remained relatively stable. Correlation analysis identified precipitation as a key factor positively affecting the α-diversity of abundant, rare, and total taxa. Conversely, AP exhibited a negative influence on α-diversity. In addition, there was a significant negative correlation between the α-diversity of abundant and total fungal taxa and soil pH, while the α-diversity of rare fungal taxa demonstrated a notable negative correlation with soil CEC.

### 4.2. Community Composition and Functions of Abundant and Rare Taxa and Impacting Factors

Regarding the relative significance of regional disparities in explaining the β-diversity of different-abundance fungal taxa, the NMDS plot illustrates dispersed distributions of soil fungal community structures across regions, indicating a certain level of variation. However, the RDA indicated that the combined impact of all environmental factors accounted for merely 14.7%, 16.4%, and 7.1% of the variations observed within abundant, rare, and total fungal taxa, respectively. This implies that no single environmental factor is capable of comprehensively explaining the regional variations in fungal communities. In conclusion, the structural differences observed among fungal communities within *P. sylvestris* var. *mongolica* forest soils across different regions are predominantly driven by interdependent soil environmental factors. Abundant fungal communities play a significant role in shaping the soil fungal community structure in *P. sylvestris* var. *mongolica* forests, which is consistent with findings from previous studies [27].

The dominant fungal phyla in the soil of the four regions in the *P. sylvestris* var. *mongolica* forest are similar in terms of abundant, rare, and total fungal taxa. However, there are slight differences in their relative abundances. The dominant phyla are mainly Basidiomycota and Ascomycota, which is consistent with the composition of fungal community structures in the soil of the four different habitats in the Hulunbuir sandy region and other desert areas [34,48]. Among them, Basidiomycota fungi have the highest abundance in the soil of bare sandy land and grassland, while Ascomycota fungi have the highest abundance in the soil of *P. sylvestris* var. *mongolica* forests. This is mainly because Ascomycota fungi evolve faster than Basidiomycota fungi, and they are more drought-resistant and radiation-resistant, making them more suitable for survival in harsh environments with lower vegetation coverage, such as bare sandy land and grassland [50,51]. Filamentous fungi, Micromonosporaceae, and Russula are common mycorrhizal fungi in the soil of *P. sylvestris* var. *mongolica* forests, and they can form symbiotic relationships with *P. sylvestris* var. *mongolica*, thus accounting for a large proportion of soil fungi [52]. However, there are significant differences in the fungal community structure at the class level between different-abundance fungal taxa. The abundant and total taxa are mainly composed of Agaricomycetes and Leotiomycetes, while rare taxa are mainly composed of Agaricomycetes and Sordariomycetes. Therefore, the dominant species of fungal communities show significant differences between regions, and a Pearson correlation analysis indicates that these fungal communities are mainly driven by different environmental factors in different regions. In addition, it is more likely due to differences in vegetation distribution, biomass, surface area, and other microhabitats in different habitats, so that plant roots can significantly influence the diversity of soil fungi [53]. The influence of plant roots on soil fungi mainly manifests itself in two aspects: On the one hand, plant roots can easily form a symbiosis with fungi, and the distribution of roots can directly cause changes in soil fungal diversity. The interaction between tree roots and ectomycorrhizal fungi is most prominent, and competition between fungi can to some extent reduce the richness of soil fungi in forests, thus resulting in certain differences in the composition of soil fungal communities in different habitats of *P. sylvestris* var. *mongolica* forests [22,54]. On the other hand, since most *P. sylvestris* var. *mongolica* forests are located in relatively arid areas with low rainfall and low nutrient content and high salt content in the soil, the metabolic processes of the fungi are limited. However, roots can indirectly affect fungal evenness by modifying environmental factors, through processes such as soil water movement, soil nutrient transformation, and energy transfer [55]. Additionally, as most of the selected *P. sylvestris* var. *mongolica* forests were artificially planted, different human activities and disturbances in different forest areas, such as land use practices and the use of chemical fertilizers, may also contribute to the variation. These assumptions need further investigation and reliable data verification, and they raise scientific questions worth exploring.

In addition, different abundances of fungal taxa play different ecological functions in soil ecosystems [56]. We predicted a wider variety of functional guilds in the rare and total taxa, but among them, populations for which trophic types could not be identified accounted for the vast majority of the soil fungal composition, suggesting that there is a lack of knowledge and in-depth exploration of the functions of these soil fungi. For functional types, the abundant and total taxa are dominated by symbiotic fungi, consisting mainly of ectomycorrhizal fungi, and rare taxa are dominated by saprophytic fungi, consisting mainly of undefined saprotrophic nutrient-forming fungi. Previous studies have shown that Sphagnum was typically an ectomycorrhizal fungus-dependent species capable of recruiting specific fungal taxa [57]. For example, the genera *Melanoleuca*, *Amphinema*, and *Tricholoma* were the indicator species of the intraroot fungi of the *P. sylvestris* var. *mongolica* forest in the Maowusu sandy land, which are able to establish a stable plant–soil feedback relationship with a community of interest formed by the host plant [58]. They improve plant tolerance to drought stress by inhibiting electrolyte infiltration, and can produce antibiotics that inhibit pathogen infestation, contributing to the healthy growth of *P. sylvestris* var. *mongolica*. Saprophytic fungi not only decompose organic matter but also inhibit the growth of pathogenic bacteria, such as *Talaromyces*, that have antagonistic effects on *Fusarium oxysporum* [59]. The higher relative abundance of saprophytic fungi during the mature forest stage of *P. sylvestris* var. *mongolica* was attributed to greater apoplastic inputs and organic matter accumulation, and the higher abundance of ectomycorrhizal fungi during the fall defoliation stage was attributed to lower photosynthesis in the host plant. Thus, changes in the functional group composition of fungi within roots largely determines the growth health of *P. sylvestris* var. *mongolica* forests.

### 4.3. Community Assembly of Abundant and Rare Taxa and Impacting Factors

Unraveling the mechanisms of subterranean microbial community construction is important for better understanding the maintenance and generation of terrestrial microbial diversity. The assembly processes can be broadly categorized into two classes: deterministic processes and stochastic processes [60]. Numerous studies have reported that deterministic processes are dominant in the construction of soil microbial communities in arid agricultural ecosystems, grassland ecosystems, and terrestrial hot spring ecosystems. These ecosystems are frequently subjected to human activities such as fertilization, tillage, and cultivation, as well as diverse geochemical gradients, resulting in significant environmental stress on the resident microorganisms. As a result, specific species which were capable of tolerating extreme conditions were the only ones preserved. In contrast to other ecosystems, our study found that stochastic processes contribute more significantly to the assembly of soil microbial communities in *P. sylvestris* var. *mongolica* forests than deterministic processes. Stochastic processes are typically considered to play a dominant role in environments with low environmental pressures, characterized by lower rates of organic matter decomposition and a reduced connectivity among water bodies, which in turn mitigates the impact of external environmental disturbances on microorganisms [42]. Studies have shown that, when environmental changes are relatively stable and insufficient to exert selective pressure on communities, the influence of stochastic processes surpasses that of deterministic processes [61]. Furthermore, the assembly processes of both the abundant and total fungal communities in different regions are quite similar, with stochastic processes accounting for 89–99.5% of the variation. In contrast, rare fungal communities are dominated by 64% stochastic processes and 36% deterministic processes, with significant differences observed between different regions. Previous studies have also reported findings similar to ours, namely, that rare microbial communities are more strongly controlled by deterministic processes than abundant communities and they exhibit higher levels of phylogenetic clustering [62]. These inconsistent results and findings may be attributed to variations in the investigated soil environments and the consideration of biogeography as a major driver of community assembly.

To delve further into the ecological mechanisms underlying community assembly processes, it is crucial to investigate the factors that balance stochastic and deterministic processes. A previous study has reported that the community assembly processes of abundant bacteria shifted from stochastic to deterministic with an increasing mean annual precipitation and mean annual air temperature in desert soils. For rare fungi, the community assembly processes shifted from deterministic to stochastic with an increasing mean annual air temperature. The study provides clear evidence that precipitation and temperature factors regulate the community assembly processes of abundant and rare microorganisms in desert soils [63]. Similar results were found in our study; precipitation and temperature were identified as the most critical environmental factors affecting the assembly processes of all fungal taxa, showing a positive correlation with abundant and rare taxa but a negative correlation with total taxa. This suggests that, in the soil ecosystem of the *P. sylvestris* var. *mongolica* forest, the increase in both precipitation and temperature leads to a transition in the ecological processes of abundant and rare fungal communities from stochasticity to homogenous selection, thereby highlighting the dominant role of deterministic processes in the assembly of these communities. In addition, previous studies on wetland and glacial soils have shown that microbial community assembly processes are related to pH, NH_4_^+^-N, and other properties of the soil. A recent cover crop study showed that assembly process was significantly influenced by soil SOC and TN content [64]. Similar results were found in this study, where factors such as soil pH, TN, SOC, AP, and CEC, in addition to precipitation and temperature, showed some negative correlations with the βNTI. These results suggest that the assembly processes of abundant and rare fungal communities can be affected by different indicators of soil properties. However, due to the upper limits of environmental factors, once the accumulation reaches a certain threshold, the βNTI values tend to cluster within the range of −2 < βNTI < +2. Nevertheless, if these factors exceed this range, βNTI values for different-abundance fungal communities can shift from −2 < βNTI < +2 to βNTI < −2 or from −2 < βNTI < +2 to βNTI > 2, indicating a transition from stochasticity to variable or homogeneous selection dominance. Hence, we propose that, in the ecosystem of *P. sylvestris* var. *mongolica* forest soil, deterministic processes gradually increase to reach a state of equilibrium alongside stochastic processes. This stochastic–deterministic community assembly process ultimately generates a diverse and abundant microbial community.

## 5. Conclusions

This study conducted a comprehensive comparison of the spatial patterns and assembly mechanisms between the abundant, rare, and total fungal communities of *P. sylvestris* var. *mongolica* forests across different regions, and it summarized how environmental factors drove the assembly processes of different fungal taxa. We found that regional differences and environmental factors have a greater impact on rare fungal taxa compared to abundant and total fungal taxa. Additionally, the α-diversity and functional guilds of rare taxa were higher than that of abundant taxa, suggesting that rare fungi play a crucial role in maintaining fungal diversity in *P. sylvestris* var. *mongolica* forests. Furthermore, spatial differences in soil fungal diversity and community structure resulted from different environmental factors. Stochastic processes play a dominant role in the assembly processes of all fungal taxa. Precipitation and temperature were the key environmental factors affecting community assembly.

## Figures and Tables

**Figure 1 microorganisms-12-00977-f001:**
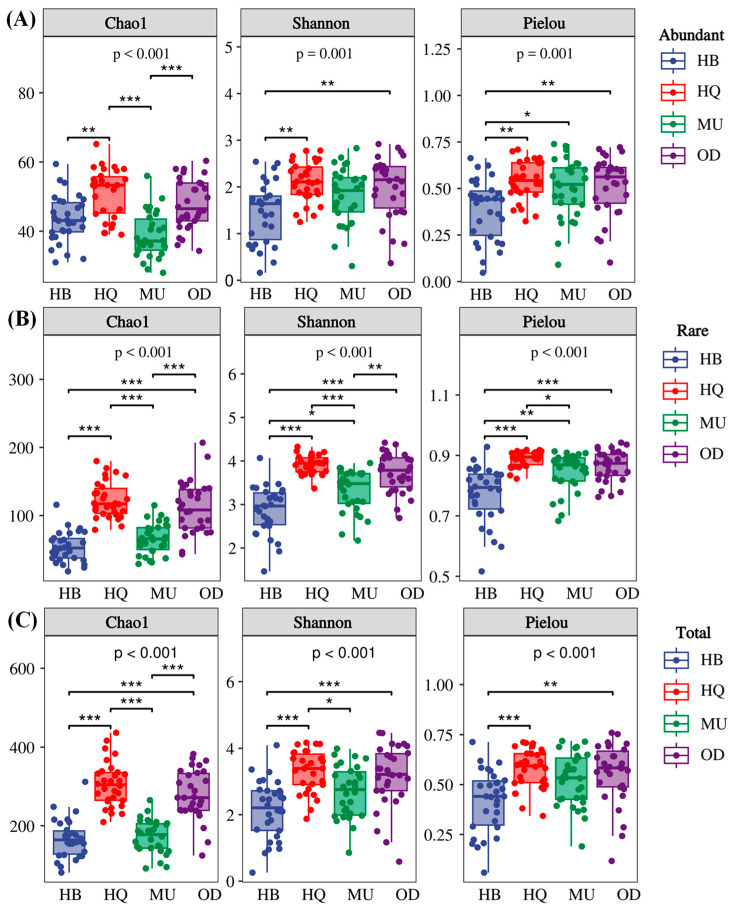
Variations of α-diversity indices of different fungal taxa across four regions. (**A**) Abundant taxa; (**B**) rare taxa; (**C**) total taxa. (* *p* < 0.05; ** *p* < 0.01; *** *p* < 0.001).

**Figure 2 microorganisms-12-00977-f002:**
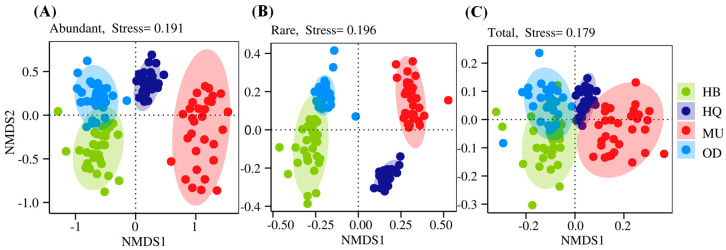
NMDS ordination of community composition of different fungal taxa across four regions. (**A**) Abundant taxa; (**B**) rare taxa; (**C**) total taxa.

**Figure 3 microorganisms-12-00977-f003:**
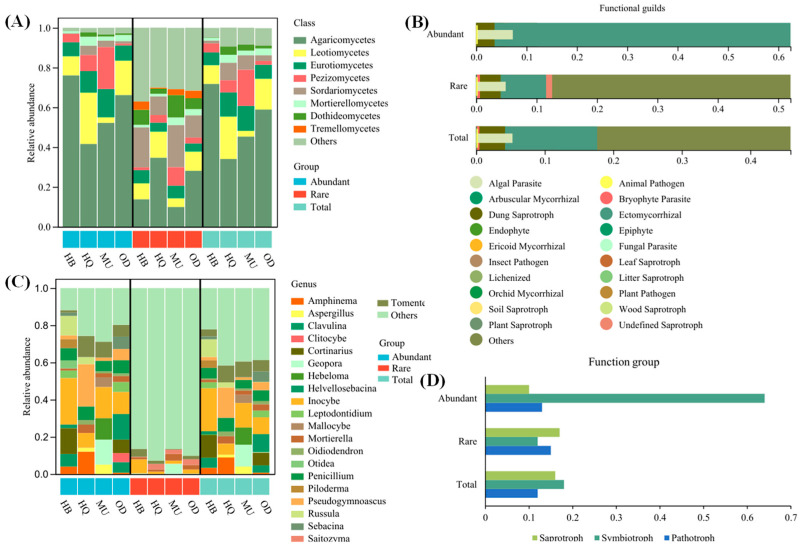
The community composition of different fungal taxa at the class (**A**) and genus levels (**C**), the functional guilds (**B**), and function group (**D**).

**Figure 4 microorganisms-12-00977-f004:**
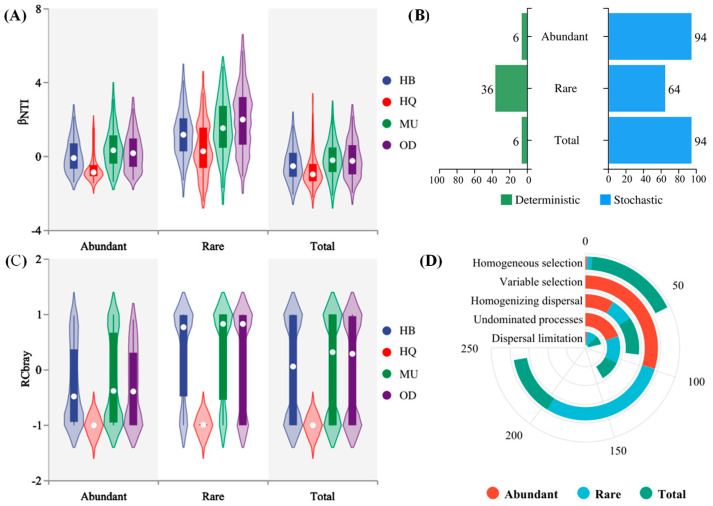
Distribution of βNTI (**A**) and RCbray (**C**) of different fungal taxa across four regions and ecological processes (**B**,**D**) shaping community assembly.

**Figure 5 microorganisms-12-00977-f005:**
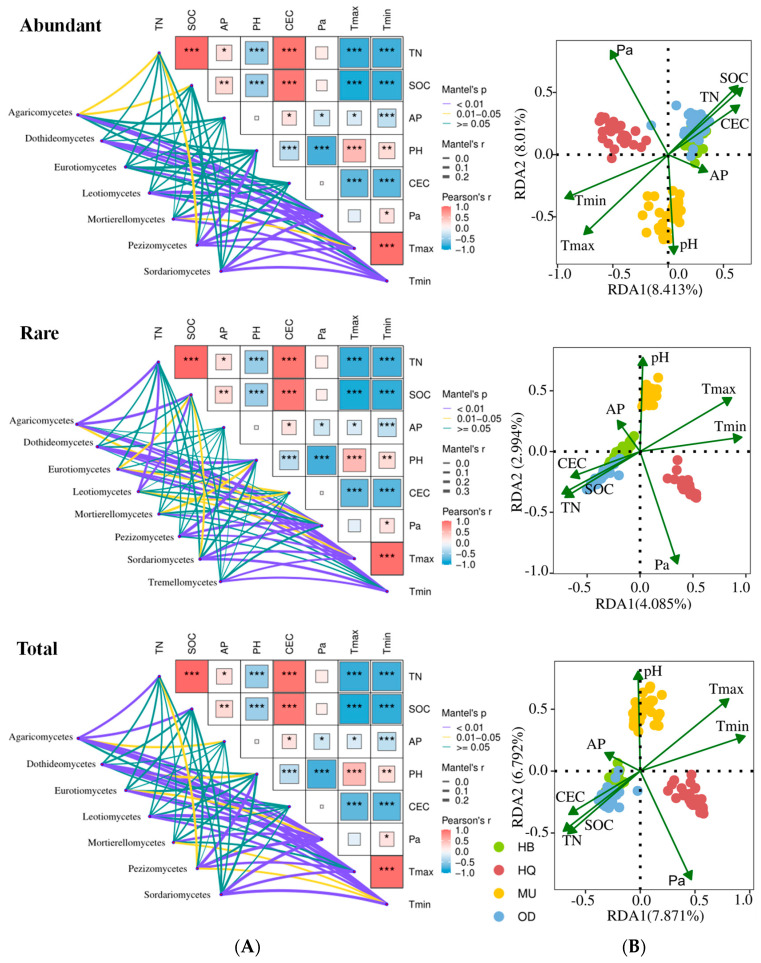
Major environmental variables in shaping the different fungal taxa based on class (**A**) and ASV (**B**) level. (**A**) Pearson’s correlation coefficients of dominant taxonomic classes and environmental factors based on Mantel tests. The edge width corresponds to Mantel’s r value, and the color of the edge indicates the statistical significance based on 999 permutations. Pairwise correlations of these variables are displayed with square size, while color gradient indicates Pearson’s correlation coefficients. Squares with asterisks represent significant levels at * *p* < 0.05; ** *p* < 0.01; *** *p* < 0.001.

**Figure 6 microorganisms-12-00977-f006:**
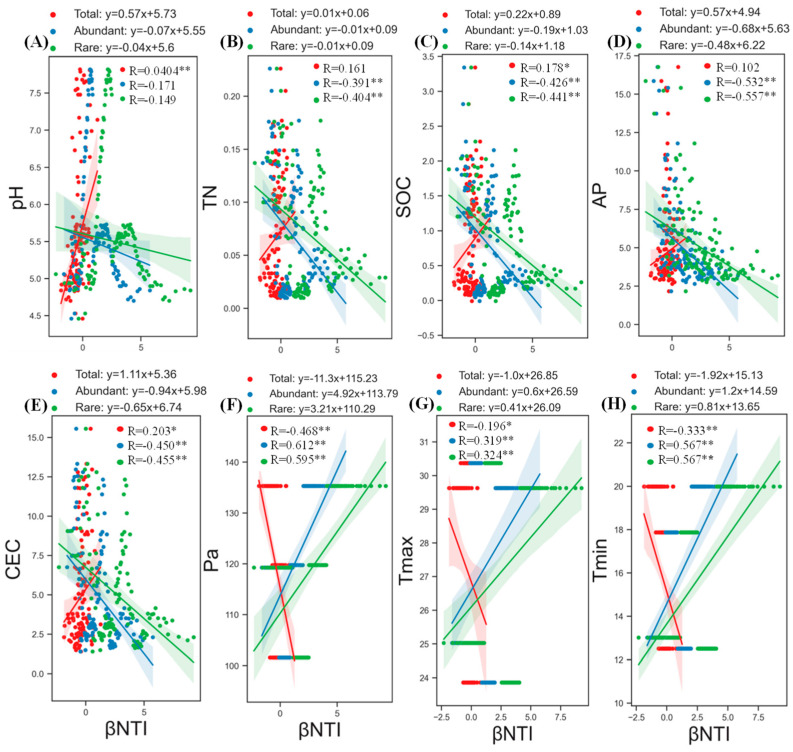
Major environmental variables (**A**–**H**) in shaping assembly processes based on βNTI. Linear regression models (different colored lines representing different fungal taxa) and associated correlation coefficients are provided on each panel. * *p* < 0.05; ** *p* < 0.01.

**Table 1 microorganisms-12-00977-t001:** General information and soil properties of sampling regions (mean values).

Site	HB	HQ	MU	OD
GC	48°14′–48°18′ N 119°58′–120°1′ E	42°42′–42°43′ N 122°29′–122°30′ E	38°18′–38°22′ N 109°40′–109°46′ E	42°22′–42°24′ N 117°15′–117°17′ E
Ele (m)	790–850	220–240	1100–1130	1520–1530
Ta (°C)	11.6	9.90	11.0	14.7
Tmax (°C)	18.4	16.6	18.6	21.9
Tmin (°C)	4.40	3.50	3.90	7.60
Pa (mm)	22.7	24.8	18.7	18.0
TN	0.11	0.04	0.01	0.10
SOC	1.35	0.42	0.16	1.39
AP	8.61	2.78	4.32	5.39
pH	5.88	6.29	8.31	6.15
CEC	8.66	2.65	2.88	5.35

GC: geographic coordinates; Ele: elevation; Ta: mean monthly temperature; Tmax: monthly maximum temperature; Tmin: monthly minimum temperature; Pa: mean monthly precipitation; TN: total nitrogen; SOC: soil organic carbon; AP: available phosphorus; pH: pH value; CEC: cation exchange capacity. HB: Honghuaerji forest park; HQ: Zhanggutai sandy forest park; MU: Yulin sandy forest park; OD: Saihanba mechanical forest farm.

**Table 2 microorganisms-12-00977-t002:** General description of different abundance of taxa data sets.

Taxa	ASV Numbers	Sequence Numbers
Abundant taxa	166 (5.8%)	3,589,112 (73.9%)
Rare taxa	2107 (73.9%)	344,539 (7.1%)
Moderate taxa	577 (20.3%)	922,869 (19%)
Total taxa	2850	4,856,520

Abundant taxa: relative abundance > 0.1%; rare taxa: relative abundance < 0.01%; moderate taxa: relative abundance 0.01~0.1%.

**Table 3 microorganisms-12-00977-t003:** Pearson correlation coefficients between α-diversity and environmental factors.

Taxa	α-Diversity	TN	SOC	AP	pH	CEC	Pa	Tmax	Tmin
Abundant	Chao1	0.143	0.114	−0.217 *	−0.483 **	0.035	0.583 **	−0.192 *	0.019
	Shannon	−0.105	−0.114	−0.195 *	−0.050	−0.135	0.159	0.122	0.200 *
	Pielou	−0.143	−0.147	−0.179	0.025	−0.155	0.079	0.169	0.219 *
Rare	Chao1	−0.066	−0.101	−0.412 **	−0.178	−0.180 *	0.472 **	0.074	0.271 **
	Shannon	−0.111	−0.135	−0.383 **	−0.059	−0.199 *	0.354 **	0.150	0.307 **
	Pielou	−0.158	−0.169	−0.290 **	−0.058	−0.225 *	0.206 *	0.211 *	0.314 **
Total	Chao1	0.022	−0.012	−0.390 **	−0.311 **	−0.107	0.582 **	0.209 *	−0.024
	Shannon	−0.092	−0.104	−0.248 **	−0.051	−0.150	0.232 *	0.215 *	0.106
	Pielou	−0.124	−0.129	−0.202 *	0.022	−0.160	0.131	0.214 *	0.143

* *p* < 0.05; ** *p* < 0.01.

## Data Availability

Data are contained within the article.

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
