# Peer review of "Spatial Distribution Patterns and Assembly Processes of Abundant and Rare Fungal Communities in *Pinus sylvestris* var. *mongolica* Forests"

_microorganisms, 2024, doi:10.3390/microorganisms12050977_

Round 1

Reviewer 1 Report

Comments and Suggestions for Authors

Line 40. “process” should read “processes”.

Line 166. “Non-multidimensional scaling” should read “Nonmetric multidimensional scaling”.

Line 179. It is not clear what “only” refers to in this sentence.

Line 199. There are seven instances of P=0 labels within the graphs. These should be changed to P<0.001.

Line 214. Fig.2. There are three instances where “Strees” should read “Stress”.

Line 214. Fig.2. The NMDS axes should be drawn with the same interval spacing. At present, an interval of 1 unit on the NMDS2 axis is much longer than the corresponding interval on the NMDS1 axis.

Line 217. Should read “functional guilds”, not “function guilds”.

Line 254. Fig. 3(C) &  3 (B) are mislabelled in the diagram, as it is 3(B) that is genus, and 3(C) that is functional guilds.

Line 255, change C to B.

Line 303. Table 3. There are 3 instances where “Richnees” needs to be replaced by “Richness”.

Line 304, footnote to Table 3.  ***, P<0.001 should be deleted, as there are no occurrences of these.

Line 339. Fig. 5. Redundancy analysis axes 1 & 2 should be drawn with the same interval spacing. At present, an interval of 0.5 on the RDA2 axis is ca. 55% longer than an interval of 0.5 on the RDA1 axis.

Line 366. Fig. 6. ***, P<0.001 should be deleted, as there are no occurrences of these.

Line 383. Replace “in was” by “has”.

Line 384. Replace “aims” by “aimed” [the study is now completed].

Line 447. What is a “mental” correlation analysis?

Line 459. Delete “so”.

Line 466. Move “worth exploring” after “questions”.

Lines 524–536. These lines repeat information that belongs in the Results section; the Discussion section should be confined to discussing the results.

Reviewer 2 Report

Comments and Suggestions for Authors

This is an interesting and overall well-presented research report.

My main concern is with the authors’ report of correlation results in Table 3. Some of the relationships that are statistically significant have relatively low actual correlation values – so low in some cases that they account for negligible variance in the variables listed in the table.

For example, in lines 297 – 302, the authors state “Other environmental factors exhibited varying types and degrees of influence on different abundance fungal taxa. For the abundant taxa, impact of pH (negative), AP (negative), and Tmax (negative), while for the rare taxa, the impact of AP (negative), Tmin (positive), and CEC (negative), and for the total taxa, impact of AP (negative), pH (negative), and Tmax (positive) were in descending order, respectively. “

In a substantial number of these cases, the correlation coefficient is in the range of R = 0.20 or lower, which means at best they are accounting for only 4% of the variance in these relations.  The authors need to recognize this limitation. It may not be statistically justifiable to try to list the results in “descending order” of value when the statistical correlations are so very low among the relationships cited.  I recommend that the authors acknowledge the limitations of this data set given the relatively low correlation coefficients, even though statistically significant. Indeed the results may be deemed significant and not attributable to chance and the null hypothesis is rejected. It is possible to achieve statistical significance with small differences, especially when the N of the data is large, but effectively the correlation values are so low as not to be particularly informative in making comparative analyses. This merely a particular point of clarification that the authors may want to address.

There are a few minor recommendations for improvement of the clarity of the text.

Lines                Recommendation

Throughout the text, the authors should use the recommended spelling for Shannon index, Pielou index and Chao1, etc.’ that is,  using capital initial letters for the names of the authors of the indices, unless the journal specifies differently.

10-11    “----assembly mechanisms of soil microorganisms is crucial to comprehend---- (The subject of the sentence is singular not plural)

30          “------- Understanding the formation of soil microbial diversity mechanisms -----”

33          “------ to explore their microbial internal community assembly mechanisms [9-10]."

50          “----- rare taxa make unequal responses to change habitat-----"

53           “ In previous studies, more research has been conducted -----”

56-57   “and rare taxa may have more important contributions-----”

65        “----at different stages of forest development (such as medium forest, near mature forest, mature forest, etc.),-----

67-68   “--- of research on the dynamics at the regional scale.”

69-70   “--- the soil of P. sylvestris var. mongolica in response to environmental changes is still unknown.”

71        “Therefore, the P. sylvestris var. mongolica forests that grew at four -----”

73        “High-throughput sequencing technology was used------”

79-80   “---- rare and total fungal community assemblies across different regions.”

80-81   “The results of this study will help to understand how the assembly of the abundant, rare and total fungal communities relate to soil biodiversity.”

92        “The basic information of the four regions is shown in Table 1.”

98        “---- through a 2-mm sieve and divided into two portions,----.

196-98 “-----, but  significant differences were found between other sampling sites (p < 0.01). The Shannon and Pielou indices of different abundances of fungal taxa differed ---------”

218      “The relative abundances were calculated on the ASVs numbers ----"

276-77 “ In general for both the rare and total taxa, ----"

299      “ ----- were further confirmed by Pearson correlation analyses.”

325      “--- (Figure 5A and Supplementary Table S1).”  Requires a space before S1.

One additional small point, I noticed in lines 89 and 90 (and lines 107-108) that the authors refer to Yulin Sandy Forest Park. However, in all geographic references that I found, the name of this particular site was presented as Yulin Sand Forest Park, not Yulin Sandy Forest Park.  The authors may want to check this.

Comments on the Quality of English Language

Overall, the text is clearly organized and there are only a few recommended changes in use of English.
